# Combing the Cancer Genome for Novel Kinase Drivers and New Therapeutic Targets

**DOI:** 10.3390/cancers11121972

**Published:** 2019-12-07

**Authors:** Pedro Torres-Ayuso, John Brognard

**Affiliations:** Laboratory of Cell and Developmental Signaling, Center for Cancer Research, National Cancer Institute, Frederick, MD 21702, USA

**Keywords:** oncogenes, kinases, precision therapies, therapy resistance, signal transduction, driver mutation

## Abstract

Protein kinases are critical regulators of signaling cascades that control cellular proliferation, growth, survival, metabolism, migration, and invasion. Deregulation of kinase activity can lead to aberrant regulation of biological processes and to the onset of diseases, including cancer. In this review, we focus on oncogenic kinases and the signaling pathways they regulate that underpin tumor development. We highlight genomic biomarker-based precision medicine intervention strategies that match kinase inhibitors alone or in combination to mutationally activated kinase drivers, as well as progress towards implementation of these treatment strategies in the clinic. We also discuss the challenges for identification of novel protein kinase cancer drivers in the genomic era.

## 1. Introduction: A Historical Perspective of Protein Kinases in Cancer—From Viral Oncogenes to an Era of Precision Medicines

Protein phosphorylation was identified at the beginning of the 20th century from studies by Levene and Alsberg, where Serine phosphorylation was mapped on cleaved vitellin [1]. However, the first protein kinase activity was not identified until the mid-1950s, when Fischer and Krebs were studying the hormonal regulation of phosphorylase b. Fischer and Krebs demonstrated that conversion of phosphorylase from its inactive form (phosphorylase b) to its active form (phosphorylase a) involved a reaction that was mediated by an enzyme that required ATP and divalent cations [2]. Using γ-[^32^P]-ATP, Fischer and Krebs showed that the reaction involved the transfer of phosphate from ATP to phosphorylase b and named the converting enzyme as phosphorylase kinase [3]. After completion of the Human Genome Project, it is now estimated that 535 protein kinases constitute the human kinome, including 50 proteins that are classified as pseudokinases due to lack of essential residues required for catalytic activity [4]. However, many pseudokinases adopt conformations that resemble the active conformation of protein kinases. In addition, certain pseudokinases bind nucleotides in a divalent cation-dependent manner, and for some, such as HER3, catalytic activity has been detected [5]. Protein kinases are classified as Ser/Thr kinase, Tyr kinases, or dual specificity kinases based on which residue they can phosphorylate. It is also known that histidine, arginine, lysine, cysteine, aspartic acid, and glutamic acid can also serve as phosphate acceptors, although phosphorylation at these residues is less studied and an exciting field for future exploration [6].

Protein kinases were not linked to cancer until the viral oncogene, v-Src, was shown to have intrinsic protein kinase activity [7]. This discovery led to the assumption that protein phosphorylation could be an important mechanism for viral transformation. Association of kinase activity with other transforming viral oncogenes was also shown for the polyomavirus middle T antigen (PymT); importantly, this kinase activity was not found in non-transforming PymT mutants, suggesting it was essential for PymT-induced transformation [8,9,10]. Work by Hunter and Sefton showed that v-Src, as well as the PymT-associated kinase, phosphorylated tyrosine residues, and led to the identification of the first tyrosine kinase [11,12]. It was not until 1983 that Courtneidge and Smith showed that the kinase associated to PymT was c-Src, the cellular homologue of v-Src [13]. By the last quarter of the 20th century, additional oncogenic viruses were identified to have intrinsic kinase activity, including v-AKT (homologue to *AKT1*), v-ABL (homologue to *ABL1*), v-ErbB (homologous to *EGFR*), and v-RAF (homologue to *RAF1*/c-RAF). Further characterization of protein kinases as components of complex signaling cascades and the emergence of techniques to modulate their expression and/or activity has been seminal to elucidate mechanisms through which certain protein kinases promote transformation. With the emergence of sophisticated DNA sequencing techniques, it has been shown that protein kinase genes are frequently mutated in tumors; some mutations lead to increased catalytic activity and provide growth and survival advantages to cancer cells [14].

Cancer genomic studies have not only aided in defining novel oncogenic mutations, but they have also contributed to the development of more effective and specific anti-tumor therapies. The concept of personalized precision medicine is based upon directly inhibiting somatically altered kinase drivers (due to indels, missense mutations, or gene fusions), where the somatic alteration confers constitutive kinase activation. Targeting these activated kinase drivers with kinase inhibitors often leads to significant reductions in tumor burden, less severe side effects compared to traditional chemotherapy, and a concomitant increase in patient survival. In fact, for many tumor types, the approval of targeted therapies for a determined group of patients has been the only significant change in disease management over the last two decades. The ever-decreasing costs associated with genomic sequencing is allowing more cases to have their tumors sequenced, translating to more patients gaining access to precision medicine-based treatments.

However, challenges have arisen and need to be addressed to ensure the full implementation of genomic-directed treatments in the clinical setting. As cancer genetic information increases due to sequencing consortiums, a primary goal is to identify the mutations in genes that act as cancer drivers (i.e., those that promote tumorigenesis), from the numerous passenger mutations (i.e., those that does not provide any growth or survival advantage) within a given tumor. Secondly, it is necessary to assess if the mutationally activated driver can be targeted with kinase inhibitors. Sequencing data indicate that, with the exception of a few driver mutations, most of the oncogenic drivers are mutated at a low frequency, remain poorly characterized, and cannot be exploited therapeutically. This has been referred as the “long right tail” of the curve of driver mutations [15,16,17]. Therefore, novel bioinformatic or functional approaches are needed to fill this gap to pinpoint additional kinase driver mutations.

Although initial benefit and long-term responses have been achieved in patients treated with kinase inhibitors, resistance to these treatments often occurs because some tumors display intrinsic mechanisms of resistance to the initial treatment, or adaptive resistance mechanisms can emerge. As discussed later, understanding potential mechanisms of tumor resistance to small molecule kinase inhibitors, and defining ways to overcome such mechanisms, is essential for the delivery of more efficacious cancer treatment combinations.

Lastly, much of past and current research in the kinase field has been dedicated to the study and development of small molecule inhibitors to target well-characterized protein kinases. Approximately 50 kinases dominate the scientific literature or have been the focus of drug development initiatives, yet genetic screens and bioinformatic analysis from cancer sequencing confirm numerous additional oncogenic kinases and suggest that several understudied protein kinases contribute to tumor progression and might represent novel actionable vulnerabilities [4,18].

## 2. Kinase Drivers: Genetic Mechanisms Leading to Increased Protein Kinase Activity

Various genetic mechanisms can result in gain of oncogenic kinase activity in cancer, including translocations, missense somatic mutations, gene amplifications, or deletion of regulatory domains that lead to increased catalytic activity (Figure 1).

### 2.1. Generating an Oncogene: Gene Fusions

Chromosome translocations can fuse two genes to create a fusion gene product with transforming properties (Figure 1A). A groundbreaking discovery was the identification of the Philadelphia chromosome (translocation t(9;22)) in chronic myelogenous leukemia (CML), a hematological cancer of which 9000 cases are diagnosed annually in the United States (US) [19]. This translocation, which is present in most CML patients, generates a fusion between the *BCR* and *ABL1* genes. The resulting gene fusion codes for a BCR–ABL1 fusion protein that has constitutive ABL1 kinase activity and can be specifically targeted with ABL inhibitors. In fact, ABL inhibitors such as imatinib showed extended benefit in patients presenting with the *BCR–ABL* fusion [20]; which led to the approval of imatinib (a.k.a., Gleevec) for the treatment of BCR–ABL-positive CML. Imatinib was one of the first small molecule kinase inhibitors approved for the treatment of cancer based on presence of a specific gene alteration.

Additional gene fusions that result in constitutive kinase activation have been detected in other cancer types [21]. For example, the *EML4–ALK* gene fusion is present in 3–5% of non-small cell lung cancer (NSCLC) cases [22]. In clinical trials, the ALK inhibitor crizotinib showed greater benefit than chemotherapy in NSCLC patients presenting with *ALK* gene rearrangement, resulting in the approval of crizotinib for the treatment of *EML4–ALK*-positive NSCLC patients (approximately 70,000 patients diagnosed annually worldwide) [23]. *ROS1* gene fusions are found in 1–2% of NSCLC cases, as well as in cholangiocarcinoma, glioblastoma, or colorectal cancer, and can be targeted with crizotinib [24]. In addition, *RET* gene fusions have been identified in NSCLC and thyroid carcinoma [25].

### 2.2. Gain-of-Function Somatic Mutations in Protein Kinase Genes

Protein kinase activity is often increased by somatic missense mutations that cause the kinase to be in the active conformation (Figure 1B). Protein kinases are molecular switches whose activation is tightly regulated and involves conformational changes between the inactive and active state, which are stabilized through several intramolecular interactions. The protein kinase catalytic domain is composed of the N-lobe, which consists of five β-sheets and a conserved α-helix termed αC-helix; and the C-lobe which consists of α-helices and loops, such as the activation loop which contains the conserved DFG motif. In the inactive state, the phenylalanine of the DFG motif sits outside of the catalytic pocket, and the αC-helix folds outside in relation with the N-lobe β-sheets. Activation of the protein kinase through binding of an allosteric regulator, or phosphorylation at the activation loop, results in a conformational change that involves the folding of the DFG motif and the αC-helix towards the catalytic pocket. The active conformation is further stabilized through the formation of the R-spine, a spatial alignment of four hydrophobic residues that form part of conserved protein kinase motifs [26,27]. Gain-of-function missense mutations in protein kinases break the equilibrium between the OFF and ON states of protein kinase by destabilizing the inactive conformation or stabilizing the active state.

For example, activating mutations in EGFR were initially found in NSCLC and generally involved mutations in exons 18–21, including in-frame deletions of exon 19, which are adjacent to the ATP binding pocket [28]. One of the most frequent EGFR activating missense mutations is the L858R substitution [29]. Leucine 858 lies at the DFG+1 position within the activation loop of EGFR, and mutation into arginine increases the level and duration of EGFR activation in response to EGF by disrupting interactions that stabilize the inactive form of the kinase [30,31]. Sustained activation of EGFR triggers downstream effectors that include the RAS–ERK and the PI3K–AKT pathways, which promote cell proliferation and survival. As with imatinib and crizotinib, patients presenting with activating EGFR mutations display significant responses to the EGFR inhibitors erlotinib or gefitinib (also known as Iressa) [29], and these have been approved for the treatment of EGFR-mutant NSCLC. Gain-of-function mutations in protein kinases genes are also frequently found in hematologic malignancies. For example, 30% of patients with acute myeloid leukemia (AML) harbor activating mutations in the FLT3 receptor tyrosine kinase (RTK). Midostaurin, a multi-targeted kinase inhibitor that targets FLT3, gained Food and Drug Administration (FDA) approval for AML patients with *FLT3* mutations in 2017 [32].

In other cases, missense mutations in protein kinases abolish the requirement for upstream regulators. For example, the substitution E17K within AKT PH domain bypasses the requirement of AKT to bind phosphatidylinositol-3,4,5-trisphosphate at the plasma membrane and renders the protein kinase constitutively active.

#### The B-RAF Oncogene

One striking case of a kinase that is hyperactivated through a missense mutation is the B-RAF (abbreviation for rapidly accelerated fibrosarcoma) kinase, which is frequently mutated in melanoma. Initially discovered as a viral oncogene, the RAF family is composed of three members: A-RAF (*ARAF*), B-RAF (*BRAF*), and C-RAF (*RAF1*). These kinases contain an RAS-binding domain and are direct effectors of the RAS GTPases, one of the most frequently mutated oncogenes in cancer [33]. RAS are small GTPases that cycle between a GTP-bound active state and a GDP-bound inactive state. Transition between these two states is coordinated by the action of guanine exchange factors (GEFs) and GTPase activating proteins (GAPs). Triggering of growth factor receptors by their ligands creates docking sites for scaffold proteins, such as Grb2, and RAS GEFs, such as SOS, that promote the exchange of GDP for GTP resulting in RAS activation. RAS GAPs counteract GEFs by enhancing RAS intrinsic GTPase activity, resulting in the hydrolysis of GTP to GDP and inactivation of RAS.

Upon activation of RAS, RAF kinases are recruited to the plasma membrane via their RAS-binding and cysteine-rich domains, which promote interaction with RAS and membrane lipids, respectively. Recruitment of RAF into the plasma membrane leads to release of autoinhibition, homo- and heterodimerization, and subsequent catalytic activation [34]. Once active, the RAF kinases phosphorylate MEK, a dual specificity kinase, which phosphorylates threonine and tyrosine residues on the ERK activation loop (Thr202, Tyr204 in ERK1), resulting in ERK activation. Active ERK phosphorylates upstream components of the cascade, including RAF, resulting in signal termination. In addition, ERK plays a critical role in the control of cell proliferation through activation of transcription factors such as ELK1, c-JUN, or c-Fos. One of the earliest genomic screens to identify mutations in genes that regulate cell proliferation and survival identified highly prevalent oncogenic missense mutations in the kinase domain of B-RAF. The most frequent mutation was the substitution V600E, which was present in approximately 50% of melanoma patients [35]. This mutation allows B-RAF to adopt a monomeric, constitutively active conformation [36,37]. Importantly, melanoma cells with this mutation were found to be more sensitive to B-RAF inhibitors, and in 2011, the FDA approved the use of vemurafenib for the treatment of B-RAF V600E-positive melanomas, making it the first drug designed using fragment-based lead discovery to gain regulatory approval [38]. However, although initial response to B-RAF inhibitors was observed, as discussed later, resistance to B-RAF inhibitors rapidly emerges.

### 2.3. Amplified Kinase Drivers

Gene amplification can lead to abnormal activation of the target gene as a result of the tumor cell harboring multiple copies of the specific oncogene (Figure 1C). Amplification of receptor tyrosine kinases from the epidermal growth factor receptor (EGFR) family members is one the best characterized examples of an amplified oncogenic kinase. Amplified *ERBB2* (17q12) is the main oncogenic driver in one third of breast cancer cases. ERBB2 (also known as HER2) is not activated by ligand binding; however, when overexpressed, it dimerizes with other EGFR family members and triggers activation of downstream signaling pathways. Additional amplified RTKs in cancer include *EGFR* (7p11.2), *FGFR1* (fibroblast growth factor receptor 1; 8p11.23), *PDGFRA* (platelet-derived growth factor receptor alpha; 4q12), *MET* (also known as HGFR, hepatocyte growth factor receptor; 7q31.2), *FLT3* (FMS-related tyrosine kinase 3; 13q12.2), and *AXL* (tyrosine protein kinase receptor UFO; 19q13.2) [28,39]. High levels of expression of ERBB2 on the cell surface led to the design and development of monoclonal antibodies directed against this amplified receptor. Trastuzumab is the first approved monoclonal antibody targeting ERBB2 for the treatment of *ERBB2*/HER2-positive breast cancers. After this discovery, additional monoclonal antibodies directed against overexpressed receptor tyrosine kinases or their ligands have been approved for the treatment of cancer, including cetuximab, approved for the treatment of head and neck squamous cell carcinoma (HNSCC) targeting EGFR and bevacizumab (Avastin), which targets the vascular endothelial growth factor A (VEGF-A), a ligand of the VEGF receptor. Importantly, these are examples of the earliest immunotherapeutic treatments against cancer, as these agents can activate the immune system, which can promote tumor cell clearance [40].

Distal amplification in the 3q chromosome (3q26-29) is one of the most prevalent genomic alterations found in cancer, mainly squamous cell carcinomas, and includes several oncogenes which are thought to act in a cooperative manner [41]. Several protein kinases within this amplicon have been characterized as genetic drivers in different malignancies, including PKCι (*PRKCI*) in NSCLC [42] and ovarian cancer [43], or LZK (*MAP3K13*) in HNSCC [44]. Additional protein kinase that are frequently amplified in cancer include FAK (*PTK2*, 8q24, in 30% of ovarian cancers), the mTORC2 activator *RICTOR* (5p13.1) in epithelial malignancies, *CDK4* (12q13-15) in sarcomas and glioblastomas, or *CDK6* (7q21) in esophageal and stomach cancers. Lastly, tandem kinase domain duplications in kinases such as EGFR, B-RAF, and FGFR1 are rare alterations (<1%) that also promote tumorigenesis [45,46,47].

## 3. Approaches for Defining New Kinase Drivers

Protein kinases are one of the most important targets for the treatment of cancer; there are currently 43 FDA-approved small-molecule kinase inhibitors for the treatment of several forms of cancer in a biomarker-specific manner [48]. Drug-discovery efforts have been directed towards a reduced number of protein kinases which are highly prevalent oncogenic drivers, such as EGFR, ERBB2, or B-RAF.

Increasingly, next generation sequencing (NGS) is guiding stratification of patients into clinical trials. Tumor sequencing consortia, such as The Cancer Genome Atlas (TCGA), are aiding in the identification of thousands of unique mutations and gene fusions across different cancer types; however, most of the emerging genetic drivers identified through NGS belong to the “long right tail” of the curve of genetic drivers [17,49]. Because of their very low mutational rate, and their unknown biological significance, more powerful bioinformatic and functional tools are required to validate the actionability of these variants of unknown significance (VUS).

Several computational approaches have been developed to predict the functional consequences of a mutation and to help to prioritize genetic drivers from passenger mutations (recently reviewed by Zhao and colleagues [50]). Such approaches are based on mutation of conserved residues in protein kinases or the effect that the mutation might have on protein stability or activity. For protein kinases for which their structure, or that of a close family member, is available, methods such as molecular dynamic simulations aid in the prediction of the functional effects of specific mutations. This approach has been used, for example, to verify gain-of-function mutations in the *ABL1* gene, which occur at a low frequency in lung adenocarcinoma [51].

Despite prediction of the impact of a certain mutation on protein kinase activity, functional characterization and validation of clinical actionability is still required. siRNA- and CRISPR-based genome-wide screenings have been seminal in the identification of genetic drivers in both solid tumors and hematologic malignancies [52,53]. These approaches allow for identification of non-mutated actionable vulnerabilities. For example, CDK9 has been identified as a dependency in hepatocellular carcinoma that cooperates with Myc to sustain cancer cell survival [54]. Similarly, Wee1 is a synthetic lethal dependency in p53-mutant HNSCC [55], and several non-mutated tyrosine kinases, such as CSF-1R or ROR1, have been identified as dependencies in leukemia patients [52]. Moreover, genome-wide screenings are being used in in vivo and preclinical cancer models that will lead to discovery of novel actionable alterations or more efficacious combination therapies [56,57]. In addition, targeted screens might aid in the validation of rare cancer variants [58]. This approach benefits from developing a small screening format in which only the genes harboring missense mutations are assessed with a defined endpoint, i.e., reduced viability after gene knockdown. In addition, targeted screenings help to reduce the large list of mutations identified in tumors which lack functional relevance, i.e., passenger mutations, and facilitate further functional validation of the selected targets in relevant cancer models.

One advantage of implementation of NGS in the oncology clinical setting is the increase of the sample size. This has been exploited for the identification of mutational hotspots in cancer drivers that are present at a very low frequency (below 1 in 1000 patients) and for which inhibitors are clinically available [17]. Another approach for functional characterization of VUS was recently developed by Ng and colleagues [49]. In their study, the authors interrogated two cancer mutation databases (TCGA and MD Anderson Cancer Center) for selection of mutations. Selected mutations were assessed functionally in an overexpression screening using two cell lines that required growth factors for survival, thus allowing the characterization of VUS as oncogenic, tumor suppressive, or non-functional. Importantly, this type of screening allowed the identification of weak drivers whose identification might not be possible using standard loss-of-function screenings which favor stronger drivers [49]. Nonetheless, weak drivers are likely to act in a cooperative manner and co-targeting approaches will be needed to achieve more complete patient responses [51].

Outside of the identification of genetic drivers, several challenges need to be addressed to benefit a larger cohort of cancer patients from precision medicine-driven oncology. Challenges include changes in the clinical trial structure to become more flexible, as well as implementation of technologies beyond DNA, such as gene expression profiling or signaling pathway analysis. DNA sequencing alone only identifies mutated genes, but these might not be expressed in the tumor cells and may represent passenger mutations. Optimization of technologies such as RNAseq or proteomic analysis, including mass spec, may be required in the clinical setting and will aid in the identification of novel actionable vulnerabilities [59,60,61].

## 4. Challenges for Targeting Kinases: Overcoming Resistance

Cancer genomic sequencing has enhanced our ability to identify kinase drivers, which has aided in the development of numerous precision therapies. The finding that patients with a specific genetic alteration in a protein kinase gene significantly benefit from inhibitors targeting that constitutively activated kinase, with fewer side effects than conventional therapies, led to a new era of genomics-driven cancer treatments [59]. However, a major hurdle to achieve the maximum potential of targeted therapies in the clinic is the emergence of resistance to protein kinase inhibitors. While protein kinase inhibitors show remarkable responses at the initial stages of treatment, resistance inevitably emerges.

Several mechanisms have been identified that promote cancer resistance to protein kinase inhibitors and are a reflection of the complex regulation of signaling cascades in a genetically unstable context (Figure 2). Resistance mechanisms to protein kinase inhibitors include mutations within the target kinase that alter the drug binding pocket size to preclude drug from binding but still allow ATP to bind, and thus preserve catalytic activity (Figure 2A). These mutations in the targeted protein kinase normally involve substitutions in or around the kinase hinge region, which lies between the N- and C-lobe of the catalytic domain. This region contains the so called “gatekeeper” residue, an amino acid within the catalytic pocket of protein kinases that interacts with the N^6^ position of ATP. Substitutions of this residue, normally by more bulky residues, weaken or impair inhibitor binding to the catalytic pocket, while retaining ATP binding and catalytic activity. This mechanism was initially identified in patients treated with tyrosine kinase inhibitors, such as gefitinib (EGFR T790M) [62] or imatinib (BCR-ABL T315I) [63,64]. In addition to mutations of the gatekeeper residue, other substitutions can happen around this amino acid that have similar consequences. These include mutations in the conserved hydrophobic gatekeeper +2 position, the conserved gatekeeper +6 glycine, or the gatekeeper +7 position. Since this is the most frequent region of mutations, it is commonly referred to as the “resistance tetrad”. Outside of the hinge region, mutations that confer resistance to small molecule inhibitors can occur in the +5 position of the G-loop, or the −/+1 positions of the DFG motif [65].

The release of negative feedback regulatory mechanisms with the consequent undesired pathway rebound or hyperactivation of compensatory signaling cascades that bypass the inhibited signaling axis are another common mechanism of resistance (Figure 2B). Signaling cascades involved in the regulation of cell proliferation, survival, differentiation, or migration are tightly controlled to maintain homeostatic activation of the biological processes they modulate. This is generally achieved through the existence of negative feedback loops, through which downstream effectors within a signaling cascade inhibit the pathway, normally through post-translational modification of upstream activators. In addition, biological functions are frequently modulated by a combination of signaling pathways that are inter-regulated, so that balanced activity of each signaling module is ensured. As a result, pharmacological inhibition of these oncogenic kinase drivers suppresses their activity and, unfortunately, also relieves negative feedback mechanisms, leading to undesired pathway rebound or hyperactivation of compensatory signaling cascades that bypass the inhibited signaling axis. This scenario has been well documented in the clinical setting, for example, with the inhibition of protein kinases such as mTOR, AKT, or RAF [66,67,68,69,70]. In general, inhibitors of these kinases lead to loss of repression of upstream RTK signaling, and therefore, the inhibitors can have the effect of activating RTK signaling. Interestingly, in the context of B-RAF V600E mutant melanoma, it has been observed that cells that apparently develop resistance against RAF or MEK inhibitors undergo massive apoptosis when the drugs are removed, i.e., drug holidays, demonstrating that cancer cells have become addicted to the inhibitor. Mechanistically, this results from deleterious hyperactivation of ERK once the inhibitor is removed and this can lead to suppression of the pathways activated that promote resistance [71]. Further understanding of the underlying mechanisms of drug addiction will guide rational alternation of different treatments to achieve stronger anti-cancer responses.

Overexpression of the target kinase or another protein with similar functionality to the inhibited target which allows signaling bypass is another common mechanism of resistance. For example, B-RAF gene amplification [72] or expression of an aberrantly spliced B-RAF [73] can confer resistance to B-RAF inhibitors. In addition, increased expression of kinases that directly phosphorylate MEK, such as the COT kinase and the MLKs (mixed lineage kinases), can directly promote resistance to RAF inhibitors (Figure 2C) [74,75].

Another mechanism of resistance is inhibitor-induced alterations of intramolecular or intermolecular interactions. This can lead to “paradoxical” re-activation of downstream signaling and has been observed for the RAF kinases (Figure 2D). As stated above, B-RAF V600E is one of the most frequent mutations found in melanoma patients. The V600E mutation allows B-RAF to be constitutively active in a monomeric state. Vemurafenib was designed and approved to treat melanoma patients with B-RAF V600E mutations; however, although marked responses were initially observed, resistance developed rapidly and presented in the form of more aggressive tumors, or development of secondary malignancies in the context of pre-existing RAS mutations [76,77,78,79]. The underlying biochemical mechanism has been extensively investigated. Following inhibition with vemurafenib, drug binding to wild-type (WT) B-RAF kinase induces a conformational change that abrogates the autoinhibited state and alters the RAF–dimer interface binding region, promoting the formation of RAF dimers, but also preventing the binding of additional inhibitor molecules to the drug-free monomer [37,73,80,81]. B-RAF V600 mutations are also present in other nonmelanoma cancers such as NSCLC and colorectal cancer. While striking responses were initially observed to vemurafenib in melanoma patients, response rates to vemurafenib in nonmelanoma tumors were very variable, for example, from 42% of patients with NSCLC vs. 4% of patients with colorectal cancer [82].

Lastly, intratumor heterogeneity contributes to therapy failure. It is well acknowledged that tumors are composed of heterogenous cellular populations that share a few dominant oncogenic drivers, usually acquired at the early stages of tumor development and referred to as truncal mutations. Additional mutations that are acquired in sub-clonal populations, referred to as the branch mutations, provide additional competitive advantages, such as enhanced proliferation. Tumor sub-clonality is wide-spread and found in tumors driven by potent oncogenes, including *EGFR* or mutant *KRAS* (recently revised by McGranahan and Swanton [83]). Moreover, it is hard to fully determine the exact extent of intratumor heterogeneity due to sampling bias and the genetic differences that occur as primary tumors evolve, metastasize, and establish new tumors at metastatic sites [84]. In addition to genetic heterogeneity, phenotypic heterogeneity is also a hallmark of cancer. For example, using a combinatorial siRNA screen, Yuan and colleagues found that, depending on the tissue of origin, mutant *KRAS* engages different effector pathways that affect their metabolic status and differentiation. As a result, tumors with the same mutation can have activation of different downstream effectors and this will impact response to treatments [85]. Importantly, anti-cancer treatments might impact the sub-clonal composition of tumors, with some subpopulations already harboring mutations that render them resistant to the anti-cancer treatment. These cells with the resistant mutation will have a competitive advantage and become the dominant clonal population in the therapy-resistant tumor. While these populations might be difficult to identify, implementation of new techniques such as serial profiling of circulating tumor cells or DNA will aid in identifying resistant sub-clonal cancer cells, and will aid in designing additional therapeutic intervention strategies aimed at targeting these cancer cells harboring drug-resistant mutations [84,86]. In addition, rapid genome re-organization or “genome chaos” has been described as a potential resistance mechanism to chemotherapy [87], and likely plays a role in resistance to kinase inhibitors; however, there are no known reports of kinase inhibitors inducing this phenomenon.

Overall, there is an abundance of mechanisms that can promote resistance to protein kinase inhibitors in cancer. One of the first ways to overcome the emergence of resistance is the development of second-generation kinase inhibitors that target both the wild-type and the inhibitor-resistant mutant allele, thus delaying or overcoming the development of resistance [88,89]. Since the emergence of resistance most frequently involves re-activation of the inhibited pathway or of secondary pathways due to the release of negative feedbacks, combinations of synergistic inhibitors have emerged as one of the most promising approaches for development of more potent anti-cancer treatments. siRNA- or shRNA-mediated gene knockdown screenings and CRISPR-mediated gene knockout screenings, coupled to next-generation sequencing, have been broadly used for identification of mechanisms of cancer resistance to several protein kinase inhibitors and, as a result, of new targetable vulnerabilities [90,91]. More recently, combinatorial approaches have been successfully implemented to identify vulnerabilities in *KRAS*-mutant cancers [92]. In addition to genetic screenings, combinatorial drug screenings are another efficient way to identify synergistic inhibitors that reduce or even lead to cancer regression. While the problem of combinatorial target identification has been overcome by recent technological advances, one major hurdle for the implementation of inhibitor combinations is the associated toxicity. However, the generation of more sophisticated pre-clinically relevant models, such as patient-derived xenografts (PDX), circulating-cell derived xenografts (CDX), and normal tissue- and tumor-derived organoids, will allow for the identification of effective combinations therapies with decreased associated toxicities. While initially these models did not allow for testing of immunotherapies, the establishment of humanized PDX models with a reconstituted immune system [93] or organoids that retain cells from the tumor microenvironment [94] might help overcome some limitations of these models.

## 5. Targeting Kinases to Modulate the Immune System: Combating at Different Fronts

Most research on protein kinases in cancer has focused on the role that these enzymes play in the control of proliferation or survival in a cell autonomous manner. It is well recognized that tumors are plastic and complex ecosystems, composed not only of malignant cells, but also the underlying extracellular matrix and a subset of “normal” cells that include different immune populations, endothelial cells, and fibroblasts. These non-transformed components of tumors constitute the so-called tumor microenvironment, and can also impact oncogenic signaling through reciprocal signaling [95,96]. The study of the molecular mechanisms that govern the crosstalk between cancer cells and their surrounding microenvironment, the communication between different components of the tumor microenvironment, and the understanding of how these connections affect tumor progression is a nascent and emerging field in cancer biology.

Immune evasion is a hallmark of cancer and is the result of mechanisms evolved by cancer cells to avoid their destruction by the immune system [97]. Immunotherapies are a class of anti-cancer treatments that aim to revert this process. However, while immunotherapies (i.e., checkpoint inhibitors) have achieved significant long-term responses in a subset of cancer patients, including melanoma and NSCLC, for other tumor types, the vast majority of patients still do not respond or develop resistance to these treatments [98,99].

Pro-oncogenic pathways can contribute to immune evasion (Figure 3). For example, the oncogenic activation of AKT and mTOR stimulates the expression of PD-L1 (program death ligand 1) on the surface of cancer cells [100]. The binding of PD-L1 to the inhibitory checkpoint receptor PD-1 transmits an inhibitory signal that results in diminished T-cell activation and, as a result, promotion of tumor growth. Monoclonal antibodies against PD-1 and PD-L1 have shown response in NSCLC patients; however, the combination of mTOR inhibitors and anti-PD-1 antibodies showed enhanced anti-tumor efficacy in pre-clinical models of NSCLC and colorectal cancer [100,101]. Similar results have been observed with the combination of B-RAF and MEK inhibitors with PD-1 blocking antibodies in melanoma [102,103].

Additional studies implicate several protein kinases that could be targeted to modulate the immune component of a tumor. The non-receptor tyrosine kinase FAK is an example. In squamous cell carcinomas (SCC), FAK localizes in the nucleus and promotes the transcription of chemokines that lead to the differentiation of T cells into T-reg, and thus, evasion of anti-tumor immunity [104,105]. FAK has also been involved in the evasion of the immune system in pancreatic ductal adenocarcinoma (PDAC) [106]. In this case, inhibition of FAK decreased the characteristic fibrosis found in PDAC, as well as the numbers of suppressive immune cells, including T-reg, myeloid-derived suppressor cells and M2 macrophages. Lastly, the authors showed that inhibition of FAK could increase the efficacy of anti-PD1 antibodies in their PDAC model [106]. Interestingly, in both cases, it was the activity of FAK in the tumor compartment driving the immunosuppressive effects. Currently, the combination of the FAK inhibitor defactinib with pembrolizumab (anti-PD1 antibody) is being tested in clinical trials (Trial ID: NCT02758587).

Cyclin-dependent kinase 4 and 6 (CDK4/6) inhibitors are another example of targets whose anti-tumor effect might be partially attributed to modulation of the immune system [107]. CDK4/6 control the transition through the G1 phase of the cell cycle and, as a result, CDK4/6 inhibitors normally induce cell cycle arrest but not apoptosis. It has been shown, however, that inhibition of CDK4/6 promotes anti-tumor immunity through several mechanisms. In breast cancer cells, inhibition of CDK4/6 enhanced tumor antigen presentation, suppressed the proliferation of regulatory T-cells (T-reg), and promoted CD8 T-cell mediated tumor regression [107]. In another study it was shown that inhibition of CDK4/6 enhanced T-cell activation through regulation of the NFAT transcription factors [108]. In both studies, the authors demonstrated that the combination of CDK4/6 inhibitors with immune checkpoint antibodies had a stronger effect that each treatment alone [107,108]. As a result of these examples, it has been proposed that combination of both targeted and immune-based therapies could synergize and provide stronger anti-cancer responses than treatment with each independently [109,110].

Targeting immune-specific kinases can also lead to tumor regressions, as shown in cancer pre-clinical models. Since tumor-associated cells are genetically more stable than cancer cells, resistance to these inhibitors might be less likely to develop. Recent examples of kinase inhibitors targeting cancer-associated immune cells include small molecule inhibitors directed towards Bruton’s tyrosine kinase (BTK) [111], colony-stimulating factor-1 receptor (CSF-1R) [112], and PI3Kgamma [113]. In the first case, Gunderson et al. showed that BTK inhibition, which impacts B cells and tumor-associated macrophages (TAMs), with the FDA-approved inhibitor ibrutinib restored T cell-mediated anti-tumor immunity in a model of PDAC and improves response to chemotherapy [111]. CSF-1R inhibitors also target TAMs and have been shown to have activity in models of glioblastoma [112], although resistance in these models has also been observed [114]. Lastly PI3Kgamma inhibitors, which also target TAMs, have shown anti-cancer responses in mouse models of HNSCC, breast cancer, and lung cancer [113]. Moreover, it has been proposed that targeting PI3Kgamma can help overcome resistance to checkpoint inhibitors [115]. As our understanding on the interaction between cancer cells and the immune system increases, additional kinases will be identified to target to modulate the immune system and promote an anti-tumor response. It is likely that this new avenue of research will provide groundbreaking therapeutic strategies in the coming years.

## 6. Future Hurdles and Challenges to Overcome

The approval of protein kinases inhibitors for the treatment of certain cancer types has been one of the most groundbreaking achievements in cancer management over the last three decades. These treatments benefit from being specific to the altered target and show reduced toxicity in comparison with traditional therapeutics such as chemotherapy. However, despite initial success, the emergence of resistance to protein kinase inhibitors has been a major hurdle to achieve the maximal benefit from these therapeutics. New approaches to target protein kinases have been developed, including PROTACs (proteolysis-targeting chimera). These are heterologous molecules that couple a pharmacophore, i.e., a protein kinase inhibitor, with an E3 ubiquitin ligase ligand. Therefore, PROTACs not only inhibit protein kinase activity, but also trigger degradation of that protein kinase [116]. As a result, PROTACs target both catalytic- and non-catalytic-dependent signaling of protein kinases and can thus be more potent compounds than catalytic inhibitors. PROTACs have been designed for a variety of oncogenic protein kinases, including BCR-ABL1 [117], FAK [118], and FLT3 [119]. With PROTACs entering clinical trials [120], a new era on targeting protein kinases in disease may begin. Lastly, increasing our understanding of tumor heterogeneity and evolution during treatment, as well as how protein kinases impact the crosstalk between cancer cells and their surrounding microenvironment, might aid in the development of more efficient therapeutic combinations.

## 7. Conclusions

The advent of next generation sequencing has enhanced our ability to identify new protein kinase targets for the development of anti-cancer treatments. Kinase-based targeted therapies have improved clinical outcomes for numerous cancer patients as they are more efficacious and less toxic than traditional treatments. However, several challenges still need to be addressed to gain the maximum benefit of targeting protein kinases to treat cancer. We have provided an overview of the challenges in delivering kinase-focused precision medicine intervention strategies for patients in the clinic, including identification of novel kinase oncogenic drivers, emergence of resistance to protein kinase inhibitors, and understanding the contribution of protein kinases to the tumor-microenvironment. Given that technologies such as proteomics, and cancer models like organoids are becoming more sophisticated, as well as new ways to target kinases, such as PROTACs, we anticipate that more efficient kinase-based therapeutics will be developed in the coming future.

## Figures and Tables

**Figure 1 cancers-11-01972-f001:**
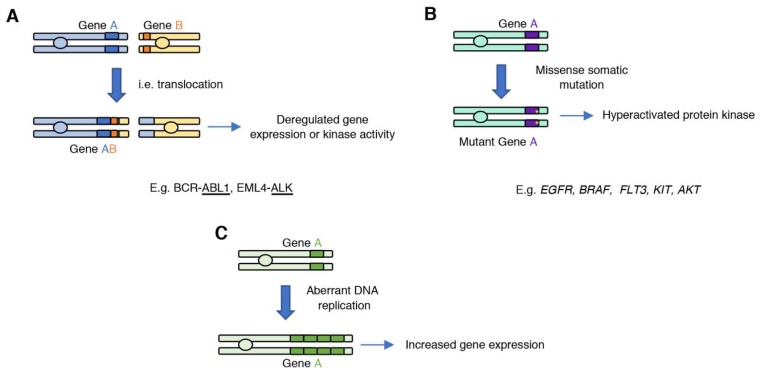
Mechanisms that can lead to the generation of oncogenic protein kinases. (**A**) Gene translocations can fuse two genes; expression or activity of the new gene product is deregulated and promotes tumorigenesis. (**B**) Introduction of somatic missense mutations during DNA replication can lead to the acquisition of gain-of-function mutations, which result in increased kinase activity of the mutant variant. (**C**) Aberrant DNA replication can lead to oncogene amplification, which can result in increased oncogene expression.

**Figure 2 cancers-11-01972-f002:**
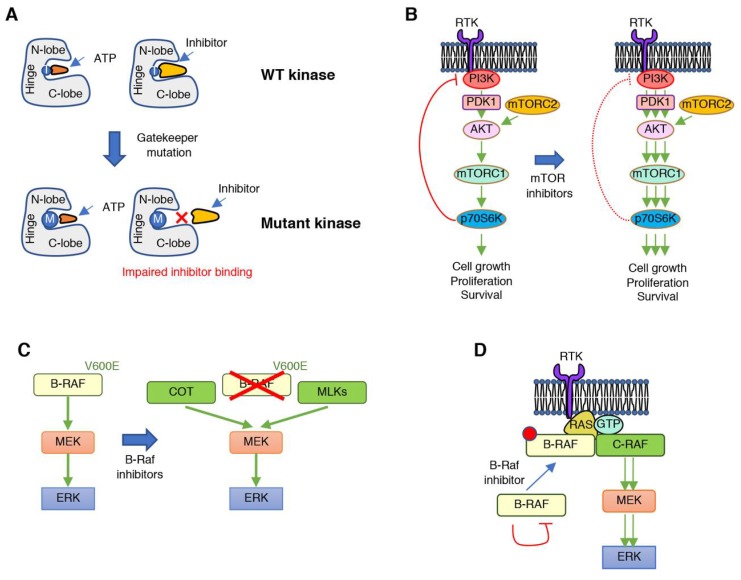
Mechanisms of tumor resistance to protein kinase inhibitors. (**A**) Mutations in the hinge region of protein kinases, such as in the gatekeeper residue (exemplified by T on the wild-type (WT) kinase) into bulkier amino acids can impair binding of small molecules due to steric restrictions. (**B**) The relief of negative regulatory signals, such as when treating with mTOR inhibitors, can lead to undesired reactivation of the inhibited pathway or of compensatory signaling cascades. (**C**) Resistance to B-RAF inhibitors is frequently achieved through increased expression of protein kinases that can phosphorylate MEK and compensate for B-RAF inhibition. (**D**) Treatment with B-RAF inhibitors can lead to paradoxical reactivation of the RAF–MEK–ERK signaling cascade by promoting B-RAF and C-RAF dimerization, as well as interaction of the RAF dimer with active RAS.

**Figure 3 cancers-11-01972-f003:**
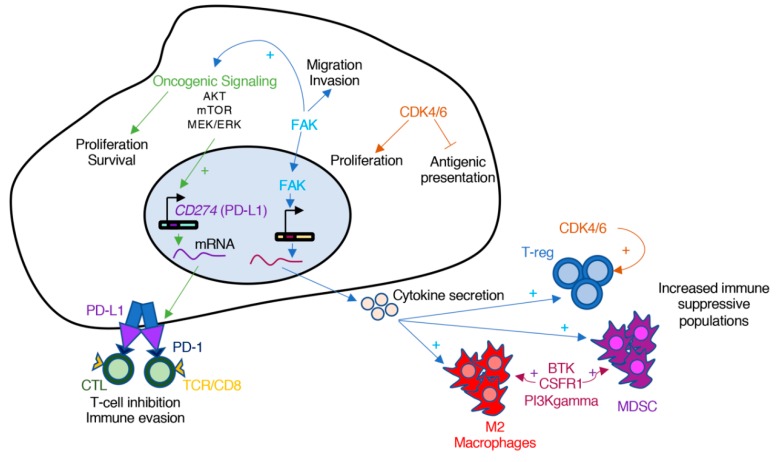
Oncogenic protein kinases can promote cancer immune evasion. Several pro-oncogenic signaling cascades are part of the mechanisms of immune evasion. Oncogenic activation of proliferation and survival axis, such as AKT, mTOR, or MEK–ERK, activate the transcription and stabilization of PD-L1, which inactivates cytolytic CD8-T cells. Other cascades such as FAK trigger the transcription of cytokines that result in increased number of immunosuppressive immune cells, including M2 Macrophages, myeloid-derived suppressor cells (MDSC), or regulatory T-cells (T-reg). Lastly, activation of protein kinases such as CDK4/6 inhibit antigen presentation by immune cells and sustain the proliferation of immunosuppressive T-reg cells.

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
