# Peer review of "Combing the Cancer Genome for Novel Kinase Drivers and New Therapeutic Targets"

_cancers, 2019, doi:10.3390/cancers11121972_

Round 1

Reviewer 1 Report

The authors highlighted a historical perspective of kinases including fusion gene products in carcinogenesis and cancer genomic studies that allow patients gaining access to effective treatments. Also, they draw attention to bioinformatic analysis from cancer sequencing in order to identify new oncogenic kinase drivers, and to challenges for overcoming resistance to the initial treatment.

This is a well-written review focusing on oncogenic protein kinases. A few minor revisions are listed below.

It is not clear for me that the authors mentioned in the abstract as follows, “we highlight biomarker-based precision medicine intervention strategies that match kinase inhibitors alone or in combination to mutationally activated kinase drivers”. The reviewer would suggest such terms as “biomarker-based precision medicine” should be more clearly defined, or reworded. The details of resistance to chemotherapy are major research challenges. A couple of paper demonstrated the early tumors accumulate sub-clonal driver mutations, such as in Kras, causing intratumor heterogeneity. For the benefit of the reader, this point needs included by adding certain statements. The reviewer would suggest some of the wording should be edited. For example, on lines 34 and 69, there are repeated words in one sentence. Some typo errors should be corrected. Line 48: PymT not PyMT, which one is correct? Finally, the reviewer is wondering if it can say: the authors stress “novel kinase drivers and new therapeutic targets” for cancer as mentioned in the title, but there seems to be little contents to support this. They covered oncogenic kinases and the signaling pathways which were already known to readers. The title should be more clearly expressed in line with the contents.

Reviewer 2 Report

The review entitled "Combing the Cancer Genome for Novel Kinase Drivers and New Therapeutic Targets" provides an overview of protein kinases as therapeutic targets in cancer highlighting the genetic mechanisms leading to increased protein kinase activity, the progress for clinical developments of kinase-targeted strategies and the challenges for overcoming tumor resistance to small molecule kinase inhibitors. This is a nice paper overall, well written and informative. Figures are accurate. It has a good and balanced choice of recent reviewed papers.  

I have only few minor suggestions:

Other gene fusions resulting in constitutive kinase activation, in addition to BCR-ABL and EML4-ALK, and corresponding inhibitors should be reported or at least the authors should refer to recent reviews reporting additional kinase fusions. Lines 195 to 199, The authors should reformulate this sentence to avoid misunderstanding, since Avastin is not directed against overexpressed RTKs but it functions by ligand trapping. In the paragraph 2.3, the Authors should also add AXL (Martinelli E et al Oncotarget 2015; Ohshima K et al Sci. Rep. 2017) as additional amplified RTK in cancer. Kinase domain duplication is another oncogenic driver in cancer that should be mentioned (i.e. Gallant JN et al  Cancer Discov 2015).

Reviewer 3 Report

The authors present a relevant and timely review of the identification, functional characterization, and resistance mechanisms associated with oncogenic kinase targets. The inclusion of historical context and specific kinase examples are useful and proportionally appropriate for each section. Below are minor comments for the authors to consider.

The specific examples given in each section encompass only solid tumors. To more broadly represent progress in this field, examples could also be included from leukemias/lymphomas as there are many such oncogenic kinase genetic events and resistance to kinase inhibitors that could be mentioned. The issue of tumor heterogeneity is briefly mentioned in concluding remarks on line 445, but should be also be specifically mentioned in Section 4 for its proposed role in resistance to therapy. Section 3, lines 252-255 – The authors list implementing gene expression profiling and signaling pathway analysis as challenges. However, I am curious whether the authors see a role for transcriptional or proteomic analyses in identifying kinase pathway targets in the clinical setting? Section 3 – it may be necessary to caution that DNA-only genomic screening may identify genes that are mutated but not expressed in a particular tumor type Lines 352-355 – should mention the limitations in xenograft and organoid models for identifying toxicities and interactions with the host immune system when screening combinatorial and novel therapies.

Reviewer 4 Report

This manuscript focuses on protein kinases and their role/deregulation in cancer. It is very well written review that provides both basic information and examples of precision therapies in cancer. Overall, the manuscript would be of interest to the readership of CANCERS. However, I think that the title is a little misleading. The authors only briefly mention about approaches for defining new kinase drivers (section 3). This part should be expanded. Several recent screenings have identified many new genetic drivers in solid tumors, as well as in lymphoma.

Additional comments:

Figure 3 is difficult to read – text/labels are barely visible.
